# A Transition Matrix-Based Extended Model for Label-Noise Learning

## Abstract

The transition matrix methods have garnered sustained attention as a class of techniques for label-noise learning due to their simplicity and statistical consistency. However, existing methods primarily focus on class-dependent noise and lack applicability for instance-dependent noise, while some methods specifically designed for instance-dependent noise tend to be relatively complex. To address this issue, we propose an extended model based on transition matrix in this paper, which preserves simplicity while extending its applicability to handle a broader range of noisy data beyond class-dependent noise. The proposed algorithm's convergence and generalization properties are theoretically analyzed under certain assumptions. Experimental evaluations conducted on various synthetic and real-world noisy datasets demonstrate significant improvements over existing transition matrix-based methods. Upon acceptance of our paper, the code will be open sourced.

## 1 Introduction

Deep neural networks have achieved remarkable success in various fields in recent years, especially in classification problems with labeled data [32, 2]. Compared to traditional methods, deep neural networks have greatly improved performance but their effects heavily depend on the accuracy of the provided labels. Bringing data with corrupted labels into the neural network model without special treatment can severely affect the prediction performance [8, 50]. However, acquiring accurately annotated data in reality can be very expensive, so a larger amount of data comes from the Internet or annotations by non-professional annotators. Therefore, it is currently worth studying and promoting how to alleviate the damage caused to the model when using noisy labels and make the model more robust, which is known as the problem of label-noise learning or called learning with noisy labels [29, 36, 10, 43, 41, 1, 35].

Various methods have been proposed for label-noise learning. Existing methods can be classified into several categories. One of them is to design novel loss functions or network structures [53, 39, 28], which reduce the impact of noisy labels to make the model more robust. Another category is sample selection based on sample loss or feature extracted, dividing samples into the clean dataset and the noisy dataset [4, 10, 13, 19]. Then they relabel the noisy labels [33, 15], or clear the noisy labels and use semi-supervised methods for learning [3, 19]. These methods are common recently and have achieved some good results. However, the process of sample selection is relatively subjective, and statistical consistency is lost after the selection, and most of them lack theoretical support. In contrast, transition matrix methods [9, 43, 22, 14, 59] have statistical consistency and usually have corresponding theoretical analysis as support, attracting continued attention and occupying an important position in various learning algorithms with label noise.

The core idea of transition matrix methods is to use a matrix measuring the transition probability from the distribution of true label to the distribution of observed noisy label. If an accurate transition matrix can be estimated and combined with observable data to obtain the noisy class-posterior probability, the distribution of clean label can be inferred for network learning. Therefore, estimating the transition matrix is the key to this type of method. However, it is infeasible to estimate an individual transition matrix for each sample without additional conditions [26]. Previous methods mostly focus on class-dependent and instance-independent label noise problems [43, 22, 51], assuming that the transition matrix is fixed for all samples. Among these methods, some [31, 43] assume the existence of anchor points to estimate the transition matrix, while other methods obtain the optimal estimation by adding a regularization term for matrix structure to weaken the anchor points assumption [22, 51]. However, these methods are not suitable for instance-dependent label noise and complex real-world data because they estimate only one matrix for all samples. Moreover, when the estimation of noisy class-posterior distribution is inaccurate, the estimation of the transition matrix may be easily affected [47], thereby affecting the estimation of the clean label distribution. Although some methods [42, 58, 52, 20] have recently been designed to use special networks or structures for instance-dependent noise situations, the estimation errors for them are still large, and the computational cost is too high to lose the concise characteristic of transition matrix methods.

Addressing the limitations of current transition matrix-based methods, this paper introduces an extended model for transition matrix that extends their applicability from class-dependent noise to a broader range of label-noise data without requiring additional techniques such as clustering or self-supervised learning. Inspired by methods that handle noise using sparse structures [57, 25], our model combines a global transition matrix with a sparse implicit regularization term [31, 25] for fitting the distribution of noisy labels across instances, replacing the need for estimating a separate transition matrix for each sample. This approach allows us to incorporate instance-level information into the model, expanding its capability beyond class-dependent noise scenarios while avoiding the unidentifiability and computational complexity of estimating instance-dependent matrices.

The structure of the following sections is as follows. In Section 2, we give relevant definitions and propose our method. In section 3 we conduct a theoretical analysis of the proposed method on a simplified model. In Section 4, we conduct experiments on various synthetic and real-world noisy datasets, comparing with other transition matrix-based methods. We conclude the paper in Section 5. In addition, we provide a more specific review of related works in Appendix A, proofs of theorems in Appendix B, and experimental details in Appendix C.

The main contributions of this paper are:

- We propose a novel extended model for transition matrix, incorporating sparse implicit regularization, which enables the extension of transition matrix methods from class-dependent noise to a broader range of noisy label data while maintaining simplicity, without the need for excessive additional framework design or sophisticated techniques.

- Under certain assumptions, we provide theoretical analysis on the convergence and generalization results of the algorithm on a simplified model. We prove the theorems proposed accordingly, giving support for the effectiveness of the proposed method.

- Our proposed method achieves significant improvements compared to previous transition matrix methods on both synthetic and real-world noisy label datasets, and produces competitive results without the need for additional auxiliary techniques.

## 2 Methodology

In this section, we give relevant definitions and propose a novel model that extends the transition matrix with implicit regularization (TMR) from class-dependent noise to more label-noise. It is a convenient and end-to-end model. We will formulate the method in detail and illustrate it theoretically.

### 2.1 Preliminaries

Let $\mathcal{X} \subset \mathbb{R}^d$ be the feature space, $\mathcal{Y} = \{1, 2, \cdots, C\}$ be the label space, where $C$ is the number of classes. Random variables $(X, Y), (X, \tilde{Y}) \in \mathcal{X} \times \mathcal{Y}$ denote the underlying data distributions with true and noisy labels respectively. In general, we can not observe the latent true data samples

87  $\mathbb{D}_{(N)} = \{(\boldsymbol{x}_i, y_i)\}_{i=1}^N$, but can only obtain the corrupted data $\tilde{\mathbb{D}}_{(N)} = \{(\boldsymbol{x}_i, \tilde{y}_i)\}_{i=1}^N$, where $\tilde{y} \in \mathcal{Y}$ is
88  the noisy label corrupted from the true label $y$, while denote corresponding one-hot label as $\boldsymbol{y}$ and $\tilde{\boldsymbol{y}}$.

89  Transition matrix methods use a matrix $\boldsymbol{T}(\boldsymbol{x}) \in [0,1]^{C \times C}$ to represent the probability from clean
90  label to noisy label, where the $ij$-th entry of the transition matrix is the probability that the instance $\boldsymbol{x}$
91  with the clean label $i$ corrupted to a noisy label $j$. The matrix satisfies the requirement that the sum
92  of each row $\sum_{j=1}^C \boldsymbol{T}_{ij}(\boldsymbol{x})$ is 1, and usually has the requirement for $\boldsymbol{T}_{ii}(\boldsymbol{x}) > \boldsymbol{T}_{ij}(\boldsymbol{x}), \forall j \neq i$. The
93  set of possible values for $\boldsymbol{T}$ is denoted as $\mathbb{T} = \left\{ \boldsymbol{T} \in [0,1]^{C \times C} | \sum_{j=1}^C \boldsymbol{T}_{ij} = 1, \boldsymbol{T}_{ii} > \boldsymbol{T}_{ij}, \forall j \neq i \right\}$.
94  Let $P(\boldsymbol{Y}|X = \boldsymbol{x}) = [P(Y = 1|X = \boldsymbol{x}), \cdots, P(Y = C|X = \boldsymbol{x})]^\top$ be the clean class-posterior
95  probability and $P(\tilde{\boldsymbol{Y}}|X = \boldsymbol{x}) = [P(\tilde{Y} = 1|X = \boldsymbol{x}), \cdots, P(\tilde{Y} = C|X = \boldsymbol{x})]^\top$ be the noisy
96  class-posterior probability, the formula can be write as:

$$P(\tilde{\boldsymbol{Y}}|X = \boldsymbol{x}) = \boldsymbol{T}(\boldsymbol{x})^\top P(\boldsymbol{Y}|X = \boldsymbol{x}). \tag{1}$$

97  Though estimating the transition matrix and the noisy class-posterior probability, the clean class-
98  posterior probability can be inferred by $P(\boldsymbol{Y}|X = \boldsymbol{x}) = \boldsymbol{T}(\boldsymbol{x})^{-\top} P(\tilde{\boldsymbol{Y}}|X = \boldsymbol{x})$, where the symbol
99  $-\top$ denotes the transpose of the inverse matrix. Alternatively, the neural network can be utilized to
100 fit the clean label distribution by the loss function:

$$\mathcal{L} = \frac{1}{N} \sum_{i=1}^N \ell\left(\boldsymbol{T}(\boldsymbol{x}_i)^\top f_{\boldsymbol{\theta}}(\boldsymbol{x}_i), \tilde{\boldsymbol{y}}_i\right), \tag{2}$$

101 where $f_{\boldsymbol{\theta}}(\cdot) : \mathcal{X} \rightarrow \Delta^{C-1}$ ($\Delta^{C-1} \subset [0,1]^C$ is the $C$-dimensional simplex) is a differentiable
102 function represented by a neural network with parameters $\boldsymbol{\theta}$ and $\ell$ is a loss function usually using
103 cross-entropy (CE) loss. Therefore, the key to addressing the problem in this class of methods lies in
104 how to estimate the transition matrix.

105 Since it is difficult to estimate the transition matrix $\boldsymbol{T}(\boldsymbol{x})$ individually for each sample, the majority
106 of existing methods [31, 10, 22] focus on studying the class-dependent and instance-independent
107 transition matrix, i.e., $\boldsymbol{T}(\boldsymbol{x}) = \boldsymbol{T}$ for $\forall \boldsymbol{x}$. However, these methods are limited by the assumption
108 of class-dependence and cannot be directly applied to instance-dependent label noise with good
109 effectiveness. Our objective is to make improvement and extension based on this limitation.

## 2.2 Transition Matrix with Implicit Regularization

111 The main issue with directly applying class-dependent transition matrix methods to instance-
112 dependent noise lies in using a fixed matrix $\boldsymbol{T}$, multiplying with clean class-posterior probability
113 $P(\boldsymbol{Y}|X)$, i.e., $\boldsymbol{T}^\top P(\boldsymbol{Y}|X)$ is not always equal to the noisy class-posterior probability $P(\tilde{\boldsymbol{Y}}|X)$,
114 even if the probability values $P(\boldsymbol{Y}|X)$ and $P(\tilde{\boldsymbol{Y}}|X)$ are correctly estimated. Therefore, for a broader
115 range of label-noise scenarios, relying solely on a fixed matrix $\boldsymbol{T}$ is insufficient.

116 The core idea of our proposed model is to introduce a residual term $\boldsymbol{r}(X)$ to fit the distribution
117 difference between $P(\tilde{\boldsymbol{Y}}|X)$ and $\boldsymbol{T}^\top P(\boldsymbol{Y}|X)$, where $\boldsymbol{r}(X)$ is a $C$-dimensional vector for each $X$.
118 It can be transformed into using $\boldsymbol{T}^\top P(\boldsymbol{Y}|X) + \boldsymbol{r}(X)$ to fit $P(\tilde{\boldsymbol{Y}}|X)$.

119 Intuitively, if an overall relatively suitable transition matrix $\boldsymbol{T}$ is applied to $\boldsymbol{T}^\top P(\boldsymbol{Y}|X)$, then the
120 difference between it and the probability $P(\tilde{\boldsymbol{Y}}|X)$ should be small. Inspired by methods that handle
121 noise using sparse structures [57, 25], we utilize a sparse structure to model the residual term $\boldsymbol{r}$.
122 Follow the works [30, 31, 25], using implicit regularization to represent sparse structures is a method
123 that facilitates updates and provides more stable learning performance. We exploit this technique
124 to model the residual term as $\boldsymbol{r}_i = \boldsymbol{u}_i \odot \boldsymbol{u}_i - \boldsymbol{v}_i \odot \boldsymbol{v}_i$ with respect to training sample $\boldsymbol{x}_i$, where
125 $\boldsymbol{u}_i, \boldsymbol{v}_i$ are all $C$-dimensional vectors and $\odot$ denotes an entry-wise Hadamard product. As usual, we
126 use a deep neural network $f_{\boldsymbol{\theta}}(\cdot)$ to learn the true label probability $\boldsymbol{y}_i$ w.r.t $\boldsymbol{x}_i$. So for the noisy label
127 probability distribution $\tilde{\boldsymbol{y}}_i$ given by the data, the model use $\boldsymbol{T}^\top f_{\boldsymbol{\theta}}(\boldsymbol{x}_i) + \boldsymbol{u}_i \odot \boldsymbol{u}_i - \boldsymbol{v}_i \odot \boldsymbol{v}_i$ to fit it.
128 Bring it into the loss function as:

$$\frac{1}{N} \sum_{i=1}^N \ell\left(\boldsymbol{T}^\top f_{\boldsymbol{\theta}}(\boldsymbol{x}_i) + \boldsymbol{u}_i \odot \boldsymbol{u}_i - \boldsymbol{v}_i \odot \boldsymbol{v}_i, \tilde{\boldsymbol{y}}_i\right). \tag{3}$$

129 Due to the potential existence of different $\boldsymbol{T}$ and $P(\boldsymbol{Y}|X = \boldsymbol{x})$ such that $P(\tilde{\boldsymbol{Y}}|X = \boldsymbol{x}) =$
130 $\boldsymbol{T}_1^\top P_1(\boldsymbol{Y}|X = \boldsymbol{x}) = \boldsymbol{T}_2^\top P_2(\boldsymbol{Y}|X = \boldsymbol{x})$, we add a regularization term of the volume of the
131 matrix $\mathrm{Vol}(\boldsymbol{T}) = \log\det(\boldsymbol{T})$ to loss function as [22] to ensure the transition matrix is identifiable.
132 The total loss function applied in our proposed method is:

$$\mathcal{L}(\boldsymbol{\theta}, \boldsymbol{T}, \{\boldsymbol{u}_i, \boldsymbol{v}_i\}_{i=1}^N) = \frac{1}{N} \sum_{i=1}^N \ell\left(\boldsymbol{T}^\top f_{\boldsymbol{\theta}}(\boldsymbol{x}_i) + \boldsymbol{u}_i \odot \boldsymbol{u}_i - \boldsymbol{v}_i \odot \boldsymbol{v}_i, \tilde{\boldsymbol{y}}_i\right) + \lambda \cdot \log\det(\boldsymbol{T}), \quad (4)$$

133 where we estimate parameters according to:

$$\hat{\boldsymbol{\theta}}, \hat{\boldsymbol{T}}, \{\hat{\boldsymbol{u}}_i, \hat{\boldsymbol{v}}_i\}_{i=1}^N = \underset{\boldsymbol{\theta}, \boldsymbol{T}, \{\boldsymbol{u}_i, \boldsymbol{v}_i\}_{i=1}^N}{\arg\min} \mathcal{L}(\boldsymbol{\theta}, \boldsymbol{T}, \{\boldsymbol{u}_i, \boldsymbol{v}_i\}_{i=1}^N). \quad (5)$$

134 We use the gradient descent method to update the parameters to be learned above. This method
135 constitutes our proposed extended **T**ransition **M**atrix model with sparse implicit **R**egularization
136 (TMR).The method steps are summarized in Algorithm 1 in Appendix B.1.

137 Through our model, the estimation of individual transition matrices for each sample is replaced
138 by the estimation of the global matrix and the sparse residual term. In this way, the number of
139 parameters for the transition matrix is reduced from $O(NC^2)$ to $O(NC)$, which greatly reduces the
140 difficulty of matrix estimation and computational consumption when $C$ is large. In addition, the
141 incorporation of sparse implicit regularization in combination with the transition matrix makes the
142 learning optimization process concise and efficient.

## 2.3 Integration with Contrastive Learning

144 To further improve the effectiveness of our approach, we first utilize contrastive learning as a pre-
145 trained feature extractor, followed by label learning. In this work, we also examine the enhancement
146 of the TMR method by incorporating the SimCLR method from contrastive learning as a feature
147 learner as pre-trained encoder, then resulting in TMR+.

# 3 Theoretical Analysis

149 In this section, we want to analyze the effectiveness of the proposed method theoretically under
150 specific conditions related to label-noise generation. However, it is difficult to give a direct analysis
151 of the deep neural network model. So we follow the theoretical analysis method of [25] to simplify
152 the proposed model and study on an approximately linear structure to demonstrate the effectiveness
153 of our proposed model.

## 3.1 Model Simplification and Convergence Analysis

155 The first to solve is the construction of an approximate simplified model for theoretical analysis of
156 our algorithm. Based on [12], we use first-order Taylor expansion to approximate the deep neural
157 network $f_{\boldsymbol{\theta}}(\cdot)$, which is highly over-parameterized:

$$f_{\boldsymbol{\theta}}(\boldsymbol{x}) \approx f_{\boldsymbol{\theta}_0}(\boldsymbol{x}) + \left(\frac{\partial f_{\boldsymbol{\theta}}^\top(\boldsymbol{x})}{\partial \boldsymbol{\theta}}\bigg|_{\boldsymbol{\theta}=\boldsymbol{\theta}_0}\right)^\top \cdot (\boldsymbol{\theta} - \boldsymbol{\theta}_0), \quad (6)$$

158 where $f_{\boldsymbol{\theta}}(\boldsymbol{x})$ is a C-dimensional vector, $\boldsymbol{\theta} \in \mathbb{R}^p$ $(p \gg N)$ denotes the parameters of the neural
159 network, $\frac{\partial f_{\boldsymbol{\theta}}^\top(\boldsymbol{x})}{\partial \boldsymbol{\theta}}\big|_{\boldsymbol{\theta}=\boldsymbol{\theta}_0}$ is a $p \times C$ matrix, $\boldsymbol{\theta}_0$ is the initialization of $\boldsymbol{\theta}$, symbol $\cdot$ represents matrix
160 multiplication. For simplicity, we drop the constant term in the derivation and abbreviate $\frac{\partial f_{\boldsymbol{\theta}}^\top(\boldsymbol{x})}{\partial \boldsymbol{\theta}}\big|_{\boldsymbol{\theta}=\boldsymbol{\theta}_0}$
161 as $\nabla_{\boldsymbol{\theta}_0} f(\boldsymbol{x})$. The approximate formula becomes:

$$f_{\boldsymbol{\theta}}(\boldsymbol{x}) \approx \nabla_{\boldsymbol{\theta}_0} f(\boldsymbol{x})^\top \cdot \boldsymbol{\theta}. \quad (7)$$

162 Through this processing, we simplify the deep neural network into an approximately linear structure,
163 and we use $f_{\boldsymbol{\theta}}(\boldsymbol{x}) = \nabla_{\boldsymbol{\theta}_0} f(\boldsymbol{x}) \cdot \boldsymbol{\theta}$ in the following theoretical analysis. We use a $N \times C$ matrix $\boldsymbol{F}$ to
164 represent the neural network predictions on the overall training dataset $\{(\boldsymbol{x}_i, y_i)\}_{i=1}^N$:

$$\boldsymbol{F} = \begin{bmatrix} f_{\boldsymbol{\theta}}^\top(\boldsymbol{x}_1) \\ \vdots \\ f_{\boldsymbol{\theta}}^\top(\boldsymbol{x}_N) \end{bmatrix}. \quad (8)$$

In order to be written in matrix form, we rewrite the formula (7) in vector expansion form:

$$f_{\boldsymbol{\theta}}^{\top}(\boldsymbol{x}) = [f_{\boldsymbol{\theta}}(\boldsymbol{x})_1, \cdots, f_{\boldsymbol{\theta}}(\boldsymbol{x})_C] = \mathrm{vec}(\nabla_{\boldsymbol{\theta}_0} f(\boldsymbol{x}))^{\top} \cdot \Theta, \tag{9}$$

where $\mathrm{vec}(\boldsymbol{A})$ denotes matrix expansion of a $m \times n$ matrix $\boldsymbol{A}$ by column vectors:

$$\mathrm{vec}(\boldsymbol{A}) = [\boldsymbol{A}_{1,1}, \cdots, \boldsymbol{A}_{m,1}, \cdots, \boldsymbol{A}_{1,n}, \cdots, \boldsymbol{A}_{m,n}]^{\top}, \tag{10}$$

and $\Theta$ is a $CP \times C$ matrix, denoting the Kronecker product of $C \times C$ identity matrix $\boldsymbol{I}_C$ with $\boldsymbol{\theta}$, i.e.,

$$\Theta = \boldsymbol{I}_C \otimes \boldsymbol{\theta} = \begin{bmatrix} \boldsymbol{\theta} & 0 & \cdots & 0 \\ 0 & \boldsymbol{\theta} & \cdots & 0 \\ \vdots & \vdots & \ddots & \vdots \\ 0 & 0 & \cdots & \boldsymbol{\theta} \end{bmatrix}_{CP \times C}. \tag{11}$$

We use a Jacobian matrix $\boldsymbol{G} \in \mathbb{R}^{N \times CP}$ to denote the partial derivatives of the network for each sample:

$$\boldsymbol{G} = \begin{bmatrix} \mathrm{vec}(\nabla_{\boldsymbol{\theta}_0} f(\boldsymbol{x}_1))^{\top} \\ \vdots \\ \mathrm{vec}(\nabla_{\boldsymbol{\theta}_0} f(\boldsymbol{x}_N))^{\top} \end{bmatrix}. \tag{12}$$

Then, an aggregate form of formula (7) is:

$$\boldsymbol{F} = \boldsymbol{G} \cdot \Theta. \tag{13}$$

Now we give a simplified model assumption that there exists an underlying ground truth parameter $\boldsymbol{\theta}_*$ such that corresponding $\boldsymbol{F}_*$ generated by equation (13) fits the true label distribution for sample. Meanwhile, there exist potentially true transition matrix $\boldsymbol{T}_*$ and sparse residual matrix $\boldsymbol{R}_* = [\boldsymbol{r}(\boldsymbol{x}_1), \cdots, \boldsymbol{r}(\boldsymbol{x}_N)]^{\top}$ made up of the residual terms $\boldsymbol{r}(\boldsymbol{x})$ for sample defined in Section 2.2. We assume that the $N \times C$ observed noisy label matrix $\tilde{\boldsymbol{Y}} = [\tilde{\boldsymbol{y}}_1, \cdots, \tilde{\boldsymbol{y}}_N]^{\top}$ is generated by:

$$\tilde{\boldsymbol{Y}} = \boldsymbol{F}_* \cdot \boldsymbol{T}_* + \boldsymbol{R}_*. \tag{14}$$

Expanded form after bringing in $\boldsymbol{G}$ and $\boldsymbol{\theta}_*$ is:

$$\tilde{\boldsymbol{Y}} = \boldsymbol{G} \cdot (\boldsymbol{I}_C \otimes \boldsymbol{\theta}_*) \cdot \boldsymbol{T}_* + \boldsymbol{R}_*. \tag{15}$$

The problem to be studied is transformed into given $\boldsymbol{G}$ and observed $\tilde{\boldsymbol{Y}}$ generated by formula (15), how to estimate the underlying $\boldsymbol{\theta}_*$, $\boldsymbol{T}_*$ and $\boldsymbol{R}_*$. At this time, our proposed loss function (4) to be optimized transforms into:

$$\mathcal{L}(\boldsymbol{\theta}, \boldsymbol{T}, \boldsymbol{U}, \boldsymbol{V}) = L\left(\boldsymbol{G} \cdot (\boldsymbol{I}_C \otimes \boldsymbol{\theta}) \cdot \boldsymbol{T} + \boldsymbol{U} \odot \boldsymbol{U} - \boldsymbol{V} \odot \boldsymbol{V}, \tilde{\boldsymbol{Y}}\right) + \lambda \cdot \log \det(\boldsymbol{T}), \tag{16}$$

where $L$ is matrix form from $\ell$ in formula (4), $\boldsymbol{U} = [\boldsymbol{u}_1, \cdots, \boldsymbol{u}_N]^{\top}$, $\boldsymbol{V} = [\boldsymbol{v}_1, \cdots, \boldsymbol{v}_N]^{\top}$, $\boldsymbol{R} = \boldsymbol{U} \odot \boldsymbol{U} - \boldsymbol{V} \odot \boldsymbol{V}$.

Intuitively, the parameters $\boldsymbol{\theta}, \boldsymbol{T}, \boldsymbol{R}$ are unidentifiable without other conditions due to the model (15) is over-parameterized. We need to add some conditional assumptions to ensure the convergence of parameters. The required conditions are summarized in the Appendix B.2, such as the low rank condition of $\boldsymbol{G}$, sparsity of $\boldsymbol{R}_*$, special small initialization setting, sufficiently scattered assumption [22] of clean class-posterior probability distribution, etc. Under these conditions, we try to analyze the effectiveness of our algorithm. For the simplicity of proof, we use square loss in formula (16), which can be analogized to cross-entropy loss. The parameter optimization problem (5) becomes:

$$\hat{\boldsymbol{\theta}}, \hat{\boldsymbol{T}}, \hat{\boldsymbol{U}}, \hat{\boldsymbol{V}} = \arg\min_{\boldsymbol{\theta}, \boldsymbol{T}, \boldsymbol{U}, \boldsymbol{V}} \frac{1}{2} \|\boldsymbol{G} \cdot (\boldsymbol{I}_C \otimes \boldsymbol{\theta}) \cdot \boldsymbol{T} + \boldsymbol{U} \odot \boldsymbol{U} - \boldsymbol{V} \odot \boldsymbol{V} - \tilde{\boldsymbol{Y}}\|_2^2 + \lambda \cdot \log \det(\boldsymbol{T}). \tag{17}$$

Based on this, the convergence result of parameters estimation is as follows:

**Theorem 3.1.** (**Convergence**) *Under the conditions in B.2, the estimated parameters $\hat{\boldsymbol{\theta}}$, $\hat{\boldsymbol{T}}$, $\hat{\boldsymbol{R}}$ for optimization problem (17) based on Algorithm 1 converge to the ground truth solution $\boldsymbol{\theta}_*$, $\boldsymbol{T}_*$, $\boldsymbol{R}_*$.*

The proof can be seen in Appendix B.3. Theorem 3.1 shows that under a simplified linear model and some conditions, one can use our proposed algorithm to obtain the consistent estimation of network parameters $\boldsymbol{\theta}_*$ applicable to learning with clean label data. At the same time, we can estimate the overall transition probability $\boldsymbol{T}_*$ from the correct label to the noisy label that we observed. Theorem 3.1 provides theoretical support for the effectiveness of our proposed method.

## 3.2 Generalization Analysis

In addition to convergence, the generalization of the proposed result is also worth exploring. It is finite to the amount of noisy label training data $\tilde{\mathbb{D}}_{(N)} = \{(\boldsymbol{x}_i, \tilde{y}_i)\}_{i=1}^N$ we can observe, which is considered to be randomly sampled from the overall infinite noisy data $\tilde{\mathbb{D}}$. We want to explore how well the parameters $\hat{\boldsymbol{\theta}}_{(N)}, \hat{\boldsymbol{T}}_{(N)}$ estimated by the proposed algorithm with finite data $\hat{\mathbb{D}}_{(N)}$ fit when applied to the overall data $\tilde{\mathbb{D}}$.

We define a function class about the data as

$$\mathcal{F} := \left\{ \ell(\boldsymbol{T}^\top f_{\boldsymbol{\theta}}(\cdot) + \boldsymbol{\gamma}(\cdot), \cdot) : \mathcal{X} \times \mathcal{Y} \to \mathbb{R}^+, \forall \boldsymbol{\theta} \in \mathbb{R}^p, \boldsymbol{T} \in \mathbb{T} \right\}, \tag{18}$$

where $\boldsymbol{\gamma}(\cdot)$ is the true residual term for each sample. Each element in $\mathcal{F}$ is a function about data sample. It is worth mentioning that the term of $\log \det(\boldsymbol{T})$ can be incorporated into the loss function $\ell$, without explicitly writing it separately for simplicity. Denote the $\epsilon$-cover of $\mathcal{F}$ as $\mathcal{N}_\mathcal{F} = \mathcal{N}(\epsilon, \mathcal{F}, \|\cdot\|_\infty)$, the average losses on $\tilde{\mathbb{D}}_{(N)}$ and $\tilde{\mathbb{D}}$ are $\mathcal{L}(\boldsymbol{\theta}_{(N)}, \boldsymbol{T}_{(N)}, \boldsymbol{R}_{(N)}; \tilde{\mathbb{D}}_{(N)})$ and $\mathcal{L}(\boldsymbol{\theta}, \boldsymbol{T}, \boldsymbol{R}; \tilde{\mathbb{D}})$ respectively. According to Theorem 3.1, for any fixed $\epsilon > 0$, there exists estimated parameters $\hat{\boldsymbol{\theta}}_{(N)}, \hat{\boldsymbol{T}}_{(N)}, \hat{\boldsymbol{R}}_{(N)}$ obtained by our algorithm such that:

$$\mathcal{L}(\hat{\boldsymbol{\theta}}_{(N)}, \hat{\boldsymbol{T}}_{(N)}, \hat{\boldsymbol{R}}_{(N)}; \tilde{\mathbb{D}}_{(N)}) \leq \mathcal{L}(\boldsymbol{\theta}_{(N)}, \boldsymbol{T}_{(N)}, \boldsymbol{R}_{(N)}^*; \tilde{\mathbb{D}}_{(N)}) + \epsilon, \forall \boldsymbol{\theta}_{(N)} \in \mathbb{R}^p, \boldsymbol{T}_{(N)} \in \mathbb{T} \tag{19}$$

where $\boldsymbol{R}_{(N)}^*$ is the true residual terms for $\tilde{\mathbb{D}}_{(N)}$. If we know the ground truth $\boldsymbol{R}_*$, we have the following result:

**Theorem 3.2.** *Suppose the loss function is bounded by $0 \leq \ell(\cdot, \cdot) \leq M$. For any $\delta > 0$, then with probability at least $1 - \delta$ we have*

$$\mathcal{L}(\hat{\boldsymbol{\theta}}_{(N)}, \hat{\boldsymbol{T}}_{(N)}, \boldsymbol{R}_*; \tilde{\mathbb{D}}) \leq \inf_{\boldsymbol{\theta} \in \mathbb{R}^p, \boldsymbol{T} \in \mathbb{T}} \mathcal{L}(\boldsymbol{\theta}, \boldsymbol{T}, \boldsymbol{R}^*; \tilde{\mathbb{D}}) + M\sqrt{\frac{\ln(2\mathcal{N}_\mathcal{F}/\delta)}{2n}} + M\sqrt{\frac{\ln(2/\delta)}{2n}} + 3\epsilon. \tag{20}$$

The proof can be found in Appendix B.4, using Theorem 2 in [48] as a reference. For any fixed $\epsilon > 0$, as $n$ continues to increase, the terms $\sqrt{\frac{\ln(2\mathcal{N}_\mathcal{F}/\delta)}{2n}}$ and $\sqrt{\frac{\ln(2/\delta)}{2n}}$ on the right side of the inequality (20) tend to 0. Since the $\epsilon$ can be arbitrarily small, the right side of the inequality (20) can be bounded. Looking back at the optimization target (17), we can find that the Theorem 3.2 states the estimators $\hat{\boldsymbol{\theta}}_{(N)}, \hat{\boldsymbol{T}}_{(N)}$ based on finite data $\tilde{\mathbb{D}}_{(N)}$ can also be applied relatively effectively to wider data $\tilde{\mathbb{D}}$ as long as they are randomly generated from the same pattern. It shows the generalization result of our algorithm, indicating that the estimation $\hat{\boldsymbol{\theta}}_{(N)}, \hat{\boldsymbol{T}}_{(N)}$ can be applied to new data and only the residual terms $\boldsymbol{R}$ need to be estimated separately.

# 4 Experiments

In this section, we present experimental findings to showcase the effectiveness of our proposed method compared to other methods. We evaluate our approach on both synthetic instance-dependent noisy datasets and real-world noisy datasets. More experimental details can be found in the Appendix C.

## 4.1 Datasets

We conduct experiments on following image classification datasets: CIFAR-10 and CIFAR-100 [16], CIFAR-10N and CIFAR-100N [40], Clothing1M [44], Webvision and ILSVRC12 [21]. Among them, CIFAR-10 and CIFAR-100 both have $32 \times 32 \times 3$ color images including 50,000 training images and 10,000 test images. CIFAR-10 has 10 classes while CIFAR-100 has 100 classes. We generate instance-dependent noisy data on CIFAR-10 and CIFAR-100 with noise rates ranging from 10% to 50%, following the same generation method as in [42]. CIFAR-10N and CIFAR-100N are manually annotated by human annotators, existing noisy labels within them. Clothing1M is a real-world dataset consisting of 1 million training images, consisting of 14 categories. WebVision contains 2.4 million images crawled from the websites using the 1,000 concepts in ImageNet ILSVRC12, but only the first 50 classes of the Google image subset are used in our experiments. For the validation set selection in our TMR method, we randomly sampled 10 samples from each observed class for each dataset to form the validation set, while the remaining samples were used for the training set.

## 4.2 Experimental Setup

We conduct the experiments using NVIDIA 3090Ti graphics cards. During the training process, we update the transition matrix using the Adam optimization method, the initialization is consistent with [22]. While the updates for other parameters are performed using the stochastic gradient descent (SGD) optimization method. More specifically, for CIFAR-10/10N, we use ResNet-18 as the backbone network with 300 epochs, batch size 128, learning rate for network is 0.05, 0.0005 for transition matrix and divided by 10 after the 30th and 60th epoch. For CIFAR-100/100N, we use ResNet-34 network with the same 300 epochs, batch size 128, while learning rate for network is 0.05, 0.0002 for transition matrix and divided by 10 after the 30th and 60th epoch. For clothing1M, we use a ResNet-50 pre-trained with 10 epochs, batch size 64, learning rate 0.002 for network, 0.0001 for transition matrix and divided by 10 after the 5th epoch. We use InceptionResNetV2 network on Webvision, with 100 epochs, batch size 32, learning rate 0.02 for network, 0.0005 for transition matrix and divided by 10 after the 30th and 60th epoch. For ILSVRC12, we directly use the model trained on Webvision, following the common setting in other papers in this field.

## 4.3 Comparison Methods

In our experiments, we included the following commonly used baseline methods for instance-dependent transition matrix estimation and comparison: (1) GCE [53], (2) Forward [31], (3) DMI [45], (4) VolMinNet [22], (5) PeerLoss [27] (6) BLTM [46], (7) PartT [42], (8) MEIDTM [6], (9) SOP [25] as an implicit regularization method for comparison, as well as state-of-the-art methods for comparison purposes: (10) Co-teaching [10], (11) ELR+ [24], (12) DivideMix [19], (13) SOP+ [25], (14) CC [54], (15) PGDF [5], (16) DISC [23].

Table 1: Test accuracy with instance-dependent noise on CIFAR-10/100.

| | CIFAR-10 | | | | |
| --- | --- | --- | --- | --- | --- |
| | IDN-10% | IDN-20% | IDN-30% | IDN-40% | IDN-50% |
| CE | 88.86±0.23 | 86.93±0.17 | 82.42±0.44 | 76.68±0.23 | 58.93±1.54 |
| GCE | 90.82±0.05 | 88.89±0.08 | 82.90±0.51 | 74.18±3.10 | 58.93±2.67 |
| Forward | 91.71±0.08 | 89.62±0.14 | 86.93±0.15 | 80.29±0.27 | 65.91±1.22 |
| DMI | 91.43±0.18 | 89.99±0.15 | 86.87±0.34 | 80.74±0.44 | 63.92±3.92 |
| VolMinNet | 89.97±0.57 | 87.01±0.64 | 83.80±0.67 | 79.52±0.83 | 61.90±1.06 |
| PeerLoss | 90.89±0.07 | 89.21±0.63 | 85.70±0.56 | 78.51±1.23 | 59.08±1.05 |
| BLTM | 90.45±0.72 | 88.14±0.66 | 84.55±0.48 | 79.71±0.95 | 63.33±2.75 |
| PartT | 90.32±0.15 | 89.33±0.70 | 85.33±1.86 | 80.59±0.41 | 64.58±2.86 |
| MEIDTM | 92.91±0.07 | 92.26±0.25 | 90.73±0.34 | 85.94±0.92 | 73.77±0.82 |
| SOP | 93.58±0.31 | 93.07±0.45 | 92.42±0.43 | 89.83±0.77 | 82.52±0.97 |
| TMR | **94.45±0.17** | **93.90±0.21** | **93.14±0.20** | **91.82±0.31** | **87.04±0.42** |
| | CIFAR-100 | | | | |
| | IDN-10% | IDN-20% | IDN-30% | IDN-40% | IDN-50% |
| CE | 66.55±0.23 | 63.94±0.51 | 61.97±1.16 | 58.70±0.56 | 56.63±0.69 |
| GCE | 69.18±0.14 | 68.35±0.33 | 66.35±0.13 | 62.09±0.09 | 56.68±0.75 |
| Forward | 67.81±0.48 | 67.23±0.29 | 65.42±0.63 | 62.18±0.26 | 58.61±0.44 |
| DMI | 67.06±0.46 | 64.72±0.64 | 62.80±1.46 | 60.24±0.63 | 56.52±1.18 |
| VolMinNet | 67.78±0.62 | 66.13±0.47 | 61.08±0.90 | 57.35±0.83 | 52.60±1.31 |
| PeerLoss | 65.64±1.07 | 63.83±0.48 | 61.64±0.67 | 58.30±0.80 | 55.41±0.28 |
| BLTM | 68.42±0.42 | 66.62±0.85 | 64.72±0.64 | 59.38±0.65 | 55.68±1.43 |
| PartT | 67.33±0.33 | 65.33±0.59 | 64.56±1.55 | 59.73±0.76 | 56.80±1.32 |
| MEIDTM | 69.88±0.45 | 69.16±0.16 | 66.76±0.30 | 63.46±0.48 | 59.18±0.16 |
| SOP | 74.09±0.52 | 73.13±0.46 | 72.14±0.46 | 68.98±0.58 | 64.24±0.86 |
| TMR | **76.96±0.25** | **75.94±0.32** | **74.87±0.45** | **72.56±0.60** | **69.85±0.56** |

## 4.4 Experimental Results on Synthetic Datasets

We primarily validated our TMR method against previous instance-based transition matrix methods on synthetic CIFAR-10/100 noise datasets. These methods mainly focus on estimating the transition matrix and do not leverage advanced self-supervised or semi-supervised techniques. We performed 5

independent runs for each experimental configuration, and the average values and standard deviations of each experiment are presented in Table 1.

The results demonstrate that our proposed TMR method outperforms other methods of the same category across various noise rates. It is evident that traditional transition matrix methods such as Forward and VolMinNet exhibit subpar performance when handling instance-dependent noise. On the other hand, specialized transition matrix methods designed for instance-dependent noise, such as ParT and MEIDTM, still show significant gaps compared to our method.

Furthermore, as the noise rates increase, the test accuracy of existing transition matrix methods significantly decline. This is particularly pronounced in the case of CIFAR-100 with 50% instance-dependent noise (IDN) data, where all transition matrix methods achieve test accuracy below 60%. In contrast, our proposed TMR method achieves a remarkable test accuracy of 69.85%, showcasing its exceptional performance. That demonstrates relatively robust performance of TMR with only a slight decrease as the noise rate increases.

It is worth mentioning that SOP [25], as a method that also applies implicit regularization based on sparsity assumptions, achieves comparable performance to our method when the noise rates are low. However, it still falls short of our method's performance. As the noise rate increases, SOP is more adversely affected by the noise due to its reliance on the sparsity assumption. In contrast, our proposed TMR method effectively estimates the overall trend by utilizing the transition matrix and combines it with sparsity, thereby demonstrating robustness even in the presence of higher noise rates. For instance, on CIFAR-10/100 with a 10% noise rate, TMR outperforms SOP by 0.87 and 2.87 percentage points, respectively. When the noise rate increases to 50%, TMR surpasses SOP by 4.52 and 5.61 percentage points, respectively. This clearly demonstrates the general effectiveness of our method in handling label noise learning across various noise rates.

Table 2: Test accuracy on CIFAR-10N and CIFAR-100N.

| | CIFAR-10N | | | | | CIFAR-100N |
| | Aggregate | Random 1 | Random 2 | Random 3 | Worst | Noisy |
|---|---|---|---|---|---|---|
| CE | 87.77±0.38 | 85.02±0.65 | 86.46±1.79 | 85.16±0.61 | 77.69±1.55 | 50.50±0.66 |
| Forward | 88.24±0.22 | 86.88±0.50 | 86.14±0.21 | 87.04±0.35 | 79.49±0.46 | 57.01±1.03 |
| Co-teaching | 91.20±0.13 | 90.33±0.13 | 90.30±0.17 | 90.15±0.18 | 83.83±0.13 | 60.37±0.27 |
| ELR+ | 94.83±0.10 | 94.43±0.41 | 94.20±0.24 | 94.34±0.22 | 91.09±1.60 | 66.72±0.07 |
| DivideMix | 95.01±0.71 | 95.16±0.19 | 94.89±0.23 | 95.03±0.20 | 92.56±0.42 | 71.13±0.48 |
| SOP+ | 95.61±0.13 | 95.28±0.13 | 95.31±0.10 | 95.39±0.11 | 93.24±0.21 | 67.81±0.23 |
| PGDF | 95.35±0.12 | 94.95±0.21 | 94.78±0.34 | 94.92±0.28 | 94.22±0.29 | 67.76±0.35 |
| **TMR+** | **96.06±0.21** | **95.96±0.17** | **95.74±0.31** | **95.88±0.14** | **94.91±0.22** | **70.31±0.28** |

## 4.5 Experimental Results on Real-world Datasets

In addition to comparing with transition matrix methods, we also enhanced our method, TMR, by incorporating SimCLR for feature learning, as TMR+. We compared TMR+ with other state-of-the-art methods on multiple real-world noisy datasets, and the results are presented in Table 2 and Table 3.

Table 3: Test accuracy on Clothing1M, Webvision and ILSVRC12.

| | Clothing1M | Webvision | ILSVRC12 |
|---|---|---|---|
| CE | 69.1 | - | - |
| Forward | 69.8 | 61.1 | 57.3 |
| Co-teaching | 69.2 | 63.6 | 61.5 |
| ELR+ | 74.81 | 77.78 | 70.29 |
| DivideMix | 74.76 | 77.32 | 75.20 |
| SOP+ | 74.98 | 77.60 | 75.29 |
| CC | 75.40 | 79.36 | 76.08 |
| PGDF | 75.19 | 81.47 | 75.45 |
| DISC | 73.72 | 80.28 | 77.44 |
| **TMR+** | **75.42** | **82.06** | **77.65** |

The results demonstrate that regardless of the type of noise labels, whether it is aggregated, random, or the worst-case scenario in CIFAR-10N, as well as in CIFAR-100N with more label categories, our method consistently achieves the best results in handling real-world noise. When dealing with large datasets like Clothing1M and complex image datasets like Webvision, TMR+ also achieves excellent results compared to to other SOTA methods like CC, PGDF and DISC.

Through extensive experiments on five real-world datasets, we demonstrate that our TMR method can significantly benefit from combining with self-supervised methods such as contrastive learning, indicating that high-quality features can greatly enhance our original TMR method. TMR is a plug-and-play model, where the feature extraction part can be unrelated to TMR itself and be replaced with other similar methods without requiring additional special handling.

Table 4: Ablation study of TMR, IR represents implicit regularization and TM represents transition matrix.

|        | CIFAR-10 | | CIFAR-100 | |
|--------|---------|---------|---------|---------|
|        | IDN-0.2 | IDN-0.4 | IDN-0.2 | IDN-0.4 |
| w/o IR | 90.25   | 83.31   | 66.09   | 62.47   |
| w/o TM | 93.36   | 89.67   | 72.78   | 68.59   |
| TMR    | **93.90** | **91.82** | **75.94** | **72.56** |

## 4.6 Ablation Study

Besides the aforementioned experiments, we conducted ablation studies on proposed TMR method to assess the importance of each component. Table 4 presents the comparative results under 20% and 40% instance-dependent noise rates, where "w/o" denotes "without", "TM" represents the transition matrix, and "IR" the represents implicit regularization. From the results, it can be observed that the absence of either IR or TM significantly affects the performance of our TMR method. Removing IR has a greater impact, particularly in the case of instance-dependent noise, resulting in a substantial decrease compared to TMR. While removing TM yields similar results on CIFAR-10 with a 20% noise rate, the difference becomes apparent when the noise rate increases to 40% or when applied to more complex datasets like CIFAR-100. These results indicate that both the transition matrix and implicit regularization term are crucial components in our model, highlighting the innovation of combining these two aspects in our method.

## 5 Conclusion

We propose an extended model for transition matrix that firstly combines it with sparse implicit regularization, enabling the extension of transition matrix methods from class-dependent noise to a broader range of noise scenarios while maintaining the simplicity of the model. The effectiveness of our method is theoretically analyzed under certain assumptions and validated through experiments on various noisy datasets. Additionally, our method can be enhanced by combining with pre-trained feature extractor such as contrastive learning, achieving state-of-the-art performance.

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

# A  Related Works

## A.1  Transition Matrix Methods

Most previous transition matrix methods focus on class-dependent label noise to simplify the estimation difficulty. Some of the early methods [31, 43, 47] usually assume the existence of anchor points and make the transition matrix identifiable by finding anchor points or approximate anchor points. To mitigate the anchor point assumption, VolMinNet [22] and TVD [51] add different forms of regularization for the transition matrix respectively to make it identifiable. While other methods [7, 17] try setting up unique network structure to estimate the transition matrix. Besides, [34, 48] utilize structures like meta-learning to estimate the transition matrix, but may require more clean data and computational consumption. Although the above methods are designed to handle class-dependent label noise, it is not suitable when encountering instance-dependent noise or real-world noisy data.

However, it is not feasible to estimate a transition matrix individually for each sample without other assumptions or multiple noisy labels [26]. In order to achieve an approximate estimation of the instance-dependent transition matrix, [9] uses an adaptation layer to estimate the transition matrix based on each sample's output, but the error is large due to the influence of the initial value. While [46] uses a separate network to estimate the transition matrix based on the Bayesian label. Some methods [42, 38, 58, 59] learn a part-dependent or group-dependent matrix through clustering, which is a compromise estimation method lies between instance-dependent and class-dependent methods. Other methods [6, 14] utilize similarity in feature space to assist transition matrix learning. Although these instance-dependent transition matrix methods achieve identifiability through special treatments, they are usually relatively complex and have larger errors, which is contrary to the convenient and simple characteristics of transition matrix methods.

## A.2  Implicit Regularization

Implicit regularization can be regarded as a statistical method for sparsity, playing the role of minimizing $L_1$ loss in sparse noise learning and being currently used in various models [55, 37, 49, 18, 56]. Among these methods, SOP [25] is the one worthy of special attention, which is related to our method. SOP also uses implicit regularization for noisy label learning, which gives a sparse representation of the residual term between prediction and observed noisy label. However, it does not take advantage of the overall transfer probability of noise and the noise sparsity assumption does not apply to high noise rates situation, so its performance on large noise rates data is relatively weak. We will compare it with our proposed method by experimental results specifically in Section 4.

# B  Algorithm and proofs

## B.1  Algorithm

The steps of our TMR algorithm are shown in detail in Algorithm 1.

## B.2  Conditions

*Condition* 1. For optimization problem (17), initialize parameters in the algorithm 1 with $\boldsymbol{u}_i = t\mathbf{1}$, $\boldsymbol{v} = t\mathbf{1}$, where $\mathbf{1}$ are vectors of all 1, $t$ is a small value scalar. There exists a given $\alpha_0 > 0$ such that the learning rates of gradient descent satisfy $lr(\boldsymbol{u}) = lr(\boldsymbol{v}) = \alpha lr(\boldsymbol{\theta})$, $\alpha < \alpha_0$.

*Condition* 2. Denote the rank of $\boldsymbol{G}$ in formula (15) as $r$, the number of sparse nonzero entries of $\boldsymbol{R}_*$ is $k$, $\boldsymbol{P}$ is the matrix of row vectors in SVD decomposition of $\boldsymbol{G}$. Define $s = \frac{N}{r} max_{1 \leq i \leq N} \|\boldsymbol{P}^\top \boldsymbol{e}_i\|_2^2$. Then $k, r, s$ satisfy $4k^2 rs < N$.

*Condition* 3. The row vectors of matrix $\boldsymbol{F}$ in formula (14) are sufficiently scattered, which is a weakened requirement of the anchor points assumption can be found in Definition 2 of [22].

## B.3  Proof of Theorem 3.1

*Proof.* Denote $\boldsymbol{Q} = (\boldsymbol{I}_C \otimes \boldsymbol{\theta}) \cdot \boldsymbol{T}$, the optimization problem in (17) can be written as:

$$\min \frac{1}{2} \|\boldsymbol{G} \cdot \boldsymbol{Q} + \boldsymbol{U} \odot \boldsymbol{U} - \boldsymbol{V} \odot \boldsymbol{V} - \tilde{\boldsymbol{Y}}\|_2^2 + \lambda \cdot \log \det(\boldsymbol{T}). \tag{21}$$

**Algorithm 1** Extended Transition Matrix Model with Sparse Implicit Regularization (TMR)

---

**Input:** Training data $\{(\boldsymbol{x}_i, \boldsymbol{y}_i)\}_{i=1}^{N}$, network $f_{\boldsymbol{\theta}}(\cdot)$, coefficient $\lambda$, learning rate $\tau_{\boldsymbol{\theta}}, \tau_{\boldsymbol{u}}, \tau_{\boldsymbol{v}}, \tau_{\boldsymbol{T}}$, batch size $m$, epoch number $E$, transition matrix update frequency $k$.

**Initialization:** Transition matrix $\boldsymbol{T}$ with an identity matrix, draw entries of $\{\boldsymbol{u}_i, \boldsymbol{v}_i\}_{i=1}^{N}$ from i.i.d. Gaussian distribution with zero-mean and s.t.d. 1e-8.

**for** $t = 1$ **to** $E$ **do**
   **for** $b = 1$ **to** $N/m$ **do**
      Get a sample batch $\mathcal{B} \subseteq \{1, \ldots, N\}$ with $|\mathcal{B}| = m$
      Calculate loss $\mathcal{L}$ by 4 with batch $\mathcal{B}$
      **for** $i$ **in** $\mathcal{B}$ **do**
         Update $\boldsymbol{u}_i \leftarrow \boldsymbol{u}_i - \tau_{\boldsymbol{u}} \cdot \partial \mathcal{L}/\partial \boldsymbol{u}_i$
         Update $\boldsymbol{v}_i \leftarrow \boldsymbol{v}_i - \tau_{\boldsymbol{v}} \cdot \partial \mathcal{L}/\partial \boldsymbol{v}_i$
      **end for**
      Update $\boldsymbol{\theta} \leftarrow \boldsymbol{\theta} - \tau_{\boldsymbol{\theta}} \cdot \partial \mathcal{L}/\partial \boldsymbol{\theta}$
      **if** $b/k$ is 0 **then**
         Update $\boldsymbol{T} \leftarrow \boldsymbol{T} - \tau_{\boldsymbol{T}} \cdot \partial \mathcal{L}/\partial \boldsymbol{T}$
      **end if**
   **end for**
**end for**

**Output:** Network parameters $\hat{\boldsymbol{\theta}}$, variables $\{\hat{\boldsymbol{u}}_i, \hat{\boldsymbol{v}}_i\}_{i=1}^{N}$ and transition matrix $\hat{\boldsymbol{T}}$.

---

Since implicit regularization can minimize the $L_1$ loss and according to Proposition 3.3 in [25], the first half of formula (21) will converge to a global solution for any fixed $\boldsymbol{T}$ under Condition 1. Furthermore, it can be converted into the following optimization problem:

$$\min_{\boldsymbol{Q}, \boldsymbol{R}} \frac{1}{2}\|\boldsymbol{Q}\|_2^2 + \beta\|\boldsymbol{R}\|_1, \quad \text{s.t.} \quad \tilde{\boldsymbol{Y}} = \boldsymbol{G} \cdot \boldsymbol{Q} + \boldsymbol{R}, \tag{22}$$

where $\beta = -\frac{\log t}{2\alpha}$ as defined in 1. When Condition 2 is true, the solution to problem (22) are $\boldsymbol{Q}_*$ and $\boldsymbol{R}_*$, where $\tilde{\boldsymbol{Y}}$ is produced by $\boldsymbol{G} \cdot \boldsymbol{Q}_* + \boldsymbol{R}_*$. This conclusion can be deduced from the analogy of Proposition 3.5 in [25]. Combining formula (15), we can get:

$$\boldsymbol{Q}_* = (\boldsymbol{I}_C \otimes \boldsymbol{\theta}_*) \cdot \boldsymbol{T}_*. \tag{23}$$

Therefore, problem (21) transform into an optimization problem with parameter $\boldsymbol{\theta}, \boldsymbol{T}$:

$$\min_{\boldsymbol{\theta}, \boldsymbol{T}} \log \det(\boldsymbol{T}), \quad \text{s.t.} \quad (\boldsymbol{I}_C \otimes \boldsymbol{\theta}) \cdot \boldsymbol{T} = \boldsymbol{Q}_*. \tag{24}$$

The above optimization problem has the same form as the optimization problem in [22], similar with Theorem 1 in this paper, under Condition 3, the solution to problem (24) is:

$$\hat{\boldsymbol{\theta}} = \boldsymbol{\theta}_*, \quad \hat{\boldsymbol{T}} = \boldsymbol{T}_*. \tag{25}$$

To sum up, when all conditions in Appendix B.2 are met, we can get the ground truth solution $\boldsymbol{\theta}_*$, the estimators by our algorithm converge to $\boldsymbol{T}_*, \boldsymbol{R}_*$ as mentioned in Theorem 3.1. $\qquad\square$

### B.4 Proof of Theorem 3.2

*Proof.* We use the inequality we use Hoeffding inequality [11] to help us complete the proof. Since $\hat{\boldsymbol{\theta}}_{(N)}, \hat{\boldsymbol{T}}_{(N)}$ are not independent of the samples, we use $\epsilon$-cover as mentioned in Section 3.2 to deal with the problem. In addition, the parameter $\boldsymbol{R}$ is omitted in the following proof for convenience and does not affect the understanding of the results.

According to the definition of $\epsilon$ covering, We can find a pair of parameters $\boldsymbol{\theta}_k, \boldsymbol{T}_k$ in the covering set such that:

$$|\ell(\boldsymbol{\theta}_k, \boldsymbol{T}_k; X, Y) - \ell(\hat{\boldsymbol{\theta}}_{(N)}, \hat{\boldsymbol{T}}_{(N)}; X, Y)| \leq \epsilon, \forall (X, Y) \in \mathcal{X} \times \mathcal{Y}. \tag{26}$$

Average the loss over samples, we have:

$$\mathcal{L}(\hat{\boldsymbol{\theta}}_{(N)}, \hat{\boldsymbol{T}}_{(N)}; \tilde{\mathbb{D}}) \leq \mathcal{L}(\boldsymbol{\theta}_k, \boldsymbol{T}_k; \tilde{\mathbb{D}}) + \epsilon. \tag{27}$$

To meet the requirement of probability $1 - \delta$ in Theorem 3.2, we take the probability value as $\delta/2\mathcal{N}_\mathcal{F}$ in Hoeffding inequality due to the randomness of $k$. Thus, with probability at least $1 - \delta/2\mathcal{N}_\mathcal{F}$,

$$\mathcal{L}(\boldsymbol{\theta}_k, \boldsymbol{T}_k; \tilde{\mathbb{D}}) \leq \mathcal{L}(\boldsymbol{\theta}_k, \boldsymbol{T}_k; \tilde{\mathbb{D}}_{(N)}) + M\sqrt{\frac{\ln(2\mathcal{N}_\mathcal{F}/\delta)}{2n}}. \tag{28}$$

By the definition of formula (26),

$$\mathcal{L}(\boldsymbol{\theta}_k, \boldsymbol{T}_k; \tilde{\mathbb{D}}_{(N)}) \leq \mathcal{L}(\hat{\boldsymbol{\theta}}_{(N)}, \hat{\boldsymbol{T}}_{(N)}; \tilde{\mathbb{D}}_{(N)}) + \epsilon. \tag{29}$$

According to the property of $\hat{\boldsymbol{\theta}}_{(N)}, \hat{\boldsymbol{T}}_{(N)}$ in formula (19), for any $\boldsymbol{\theta} \in \mathbb{R}^p, \boldsymbol{T} \in \mathbb{T}$,

$$\mathcal{L}(\hat{\boldsymbol{\theta}}_{(N)}, \hat{\boldsymbol{T}}_{(N)}; \tilde{\mathbb{D}}_{(N)}) \leq \mathcal{L}(\boldsymbol{\theta}, \boldsymbol{T}; \tilde{\mathbb{D}}_{(N)}) + \epsilon. \tag{30}$$

Using the Hoeffding inequality again with probability $\delta/2$, with probability at least $1 - \delta/2$ we have:

$$\mathcal{L}(\boldsymbol{\theta}, \boldsymbol{T}; \tilde{\mathbb{D}}_{(N)}) \leq \mathcal{L}(\boldsymbol{\theta}, \boldsymbol{T}; \tilde{\mathbb{D}}) + M\sqrt{\frac{\ln(2/\delta)}{2n}}. \tag{31}$$

Combining inequalities (27), (28), (29), (30), (31) and adding the probability values, we get the conclusion that with probability at least $1 - \delta$,

$$\mathcal{L}(\hat{\boldsymbol{\theta}}_{(N)}, \hat{\boldsymbol{T}}_{(N)}; \tilde{\mathbb{D}}) \leq \mathcal{L}(\boldsymbol{\theta}, \boldsymbol{T}, ; \tilde{\mathbb{D}}) + M\sqrt{\frac{\ln(2\mathcal{N}_\mathcal{F}/\delta)}{2n}} + M\sqrt{\frac{\ln(2/\delta)}{2n}} + 3\epsilon, \forall \boldsymbol{\theta} \in \mathbb{R}^p, \boldsymbol{T} \in \mathbb{T}. \tag{32}$$

$\square$

# C    Experiment details

## C.1    Experimental Setup

We conduct experiments on a single NVIDIA 3090Ti graphics card. For software, we use Python 3.11 and PyTorch 1.10 to build the models. Throughout the training process, transition matrix updates are carried out using the Adam optimization method, while updates for other parameters are performed using the stochastic gradient descent (SGD) optimization method. The experimental setup involves a few training hyper-parameters, including the backbone network used, batch size, learning rate for parameters, and weight of the regularization term. For specific experimental configurations, please refer to Table 5 in Appendix C.2.

## C.2    Hyper-parameters Setting

The backbone network and hyper-parameters of the experiments on each dataset are listed in the table 5.

Table 5: Hyper-parameters on CIFAR-10/100, Clothing-1M and Webvision.

|  | CIFAR-10 | CIFAR-100 | Clothing1M | Webvision |
|---|---|---|---|---|
| Network | ResNet18 | ResNet34 | ResNet-50 | InceptionResNetV2 |
| Batch size | 128 | 128 | 64 | 32 |
| Training samples | 50,000 | 50,000 | 1,000,000 | 65,944 |
| Epochs | 300 | 300 | 10 | 100 |
| Learning rate(lr) for network | 0.05 | 0.05 | 0.002 | 0.02 |
| lr decay for network | Cosine | Cosine | 5th | 50th |
| Weight decay for network | 5e-4 | 5e-4 | 1e-3 | 5e-4 |
| lr for $\boldsymbol{T}$ | 0.0005 | 0.0002 | 0.0001 | 0.0005 |
| lr decay for $\boldsymbol{T}$ | 30th, 60th | 30th, 60th | 5th | 50th |
| Initialization for $\boldsymbol{T}$ | -2 | -4.5 | -2.5 | -4 |
| lr for $\boldsymbol{u}, \boldsymbol{v}$ | 10, 10 | 1, 100 | 0.1, 1 | 0.1, 1 |
| lr decay for $\boldsymbol{u}, \boldsymbol{v}$ | Cosine | Cosine | 5th | 50th |
| Coefficient $\lambda$ | 0.001 | 0.001 | 0.001 | 0.001 |

Table 6: Test accuracy with symmetric and flip noise on CIFAR-10/100.

| | CIFAR-10 | | | |
|---|---|---|---|---|
| | Symmetric | | Flip | |
| | 20% | 50% | 20% | 45% |
| CE | 85.68±0.18 | 77.35±0.21 | 86.32±0.16 | 75.22±0.43 |
| GCE | 87.83±0.54 | 79.54±0.23 | 89.75±1.53 | 75.75±0.36 |
| Forward | 85.20±0.80 | 74.82±0.78 | 88.21±0.48 | 77.44±6.89 |
| DMI | 87.54±0.20 | 82.68±0.21 | 89.89±0.45 | 73.15±7.31 |
| VolMinNet | 89.58±0.26 | 83.37±0.25 | 90.37±0.30 | 88.54±0.21 |
| PeerLoss | 87.97±0.33 | 81.06±0.47 | 89.11±0.42 | 76.89±1.83 |
| BLTM | 88.30±0.38 | 82.04±0.29 | 90.77±0.45 | 80.53±1.51 |
| PartT | 89.97±0.36 | 83.72±0.56 | 90.81±0.43 | 86.15±0.87 |
| MEIDTM | 90.89±0.20 | 84.61±0.39 | 91.01±0.19 | 88.45±1.07 |
| SOP | 93.18±0.57 | 88.98±0.43 | 94.02±0.30 | 89.58±0.86 |
| TMR | **94.36±0.22** | **91.63±0.30** | **94.55±0.19** | **93.17±0.53** |
| | CIFAR-100 | | | |
| | Symmetric | | Flip | |
| | 20% | 50% | 20% | 45% |
| CE | 51.43±0.58 | 41.31±0.67 | 53.19±0.42 | 40.56±0.89 |
| GCE | 63.22±0.45 | 53.16±0.72 | 64.15±0.44 | 40.58±0.49 |
| Forward | 54.90±0.74 | 41.85±0.71 | 56.12±0.54 | 36.88±2.32 |
| DMI | 62.65±0.39 | 52.42±0.64 | 59.56±0.73 | 38.17±2.02 |
| VolMinNet | 64.94±0.40 | 53.89±1.26 | 68.45±0.69 | 58.90±0.89 |
| PeerLoss | 62.92±0.48 | 50.25±0.52 | 64.14±0.39 | 43.53±0.75 |
| BLTM | 63.46±0.58 | 52.43±0.47 | 67.10±0.22 | 48.68±0.77 |
| PartT | 65.76±0.28 | 54.88±0.93 | 69.40±0.39 | 56.12±0.61 |
| MEIDTM | 66.90±0.32 | 57.24±1.01 | 70.16±0.52 | 58.53±0.50 |
| SOP | 74.42±0.42 | 66.46±0.65 | 73.93±0.55 | 63.32±0.87 |
| TMR | **76.20±0.24** | **71.53±0.41** | **76.53±0.22** | **70.96±0.52** |

## C.3 Supplementary experiments on class-dependent noise

In addition to conducting experiments on instance-dependent noisy data, we further evaluated the general effectiveness of our method compared to other approaches by introducing class-dependent scenarios on CIFAR-10/100 datasets. Table 6 presents the comparative results on CIFAR-10/100 datasets with symmetric noise rates of 20% and 50%, as well as flip noise rates of 20% and 45%. It can be observed that for class-dependent noise, which serves as a simplified case of instance-dependent noise, our proposed method TMR outperforms other comparative methods, including transition matrix methods specifically designed for class-dependent noise, such as VolMinNet. Specifically, the transition matrix methods specifically designed for handling instance-dependent noise, such as BLTM, PartT and MEIDTM, do not show significant improvements when applied to class-dependent noise scenarios compared to the transition matrix methods designed only for class-dependent noise, such as VolMinNet. However, our proposed method, TMR, achieves significant improvements even when applied to class-dependent noise scenarios compared to VolMinNet. This indicates that our method has universal applicability and yields favorable results in both class-dependent and instance-dependent noise scenarios.

