# OpenReview forum: "A Transition Matrix-Based Extended Model for Label-Noise Learning"
_NeurIPS.cc/2024/Conference — Submitted to NeurIPS 2024_

### Official Review · Reviewer_2B3S · 2024-06-26

**Soundness:** 3
**Presentation:** 2
**Contribution:** 2
**Rating:** 3
**Confidence:** 4

**Summary:**

This paper studies the problem of learning with noisy labels. To handle the instance-dependent noise, the authors propose an extended model for transition matrix-based methods. Specifically, their model combines a class-dependent transition matrix with a sparse implicit regularization term. The authors provide a theoretical analysis of the proposed method. Experiments conducted on both synthetic and real-world noisy label datasets verify the effectiveness of their method.

**Strengths:**

1. Theoretical analysis of the convergence and generalization are provided.
2. Experiments are conducted thoroughly, including experiments on synthetic and real-world datasets. The ablation study is also conducted.

**Weaknesses:**

1. The method proposed in this paper appears to be a straightforward combination of VolMinNet and SOP.
2. The experimental results for TMR are missing for the CIFAR-N, Clothing1M, and WebVision datasets.
3. An important baseline, CCR [1], which is the state-of-the-art among transition matrix-based methods, is absent.
4. The paper lacks an analysis of the estimation error of the transition matrix. It would be beneficial to compare the estimation errors of the transition matrix for TMR against those of other baselines.

**Reference**

[1] Cheng, De, et al. "Class-dependent label-noise learning with cycle-consistency regularization." *Advances in Neural Information Processing Systems* 35 (2022): 11104-11116.

**Questions:**

1. Why is the residual term $r_i$ designed as $r_i = u_i \odot u_i - v_i \odot v_i$? If the residual term $r_i$ were designed as $r_i = u_i$, would there be any theoretical or empirical differences?
2. How does the residual term impact the estimation error of the transition matrix? Does it reduce or increase the error? An analysis or experimental results highlighting the effect of the residual term on the estimation error would strengthen the paper.

**Limitations:**

I did not find that the authors have discussed the limitations and potential negative societal impact of their work. To improve the paper, the authors can provide a thorough analysis of the limitations of their method in an independent section. For example, scenarios where the method might not perform well can be included.

---

> ### Comment · Reviewer_2B3S · 2024-08-12
>
> Since the authors did not post their responses, I maintain my initial rating.

---

### Official Review · Reviewer_T3zW · 2024-07-07

**Soundness:** 3
**Presentation:** 2
**Contribution:** 3
**Rating:** 5
**Confidence:** 3

**Summary:**

In learning from noisy labels, existing methods generally focus on class-dependent (but instance-independent) noise that can be modeled by a transition matrix $\mathbf{T}$. Some methods have also been proposed for instance-dependent noise (modeled by $\mathbf{T}(x)$). This work belongs to the latter. In particular, it proposes to implicitly model $\mathbf{T}(x)$ using an extended model based on the transition matrix $\mathbf{T}$ and a residual term $\mathbf{r}(x)$. Some theoretical properties (e.g. convergence and generalization) of the proposed algorithm (TMR) are analyzed under certain conditions. Experiments show that the proposed algorithm outperforms baselines.

**Strengths:**

**Originality**

The paper studies the challenging problem of instance-dependent label noise, which is less addressed in the literature compared to class-dependent noise. The proposed extended model for transition matrix, which is a combination of a transition matrix with residual terms, seems novel and effective. Related work is adequately cited.

**Quality**

The experiments are quite comprehensive. The paper compares the proposed method with multiple methods (including some state-of-the-art ones) on various datasets. The experimental results show that the proposed method outperforms all those baselines. Some theoretical properties (e.g. convergence and generalization) of the proposed algorithm are also analyzed under certain conditions.

**Clarity**

The description of the proposed method is clear. The experiment section is generally clearly written and well-organized.

**Significance**

The proposed method shows significant improvements compared with various baselines. Therefore, it has the potential to be adopted by other researchers and practitioners, advancing the state of the art in learning from noisy labels.

**Weaknesses:**

**Originality**

- In Lines 120-125, the residual term $\mathbf{r}(x)$ is introduced. However, it is not clear to me how novel it is compared to the previous work [57,25,30,31]. The authors should elaborate on this point.
- I can see why residual term $\mathbf{r}(x)$ might be useful, but why is it modeled as in the form in Line 124? The motivation should be explained.

**Quality**

- The convergence analysis seems very restrictive to me because it requires too many assumptions (Lines 171-175, Lines 183-186, and Appendix B.2).
- The generalization analysis (Theorem 3.2) is w.r.t. the training loss (surrogate loss) under the noisy distribution $\tilde{\mathbb D}$, but the test accuracy under the clean distribution $\mathbb D$ is what people really care about. Is it possible to prove any consistency guarantees?
- Knowledge of the ground truth $R_*$ is required to derive Theorem 3.2, but we do not know $R_*$ in practice.
- Section 3 is not clearly written, and I found it hard to follow and assess its correctness (see below).

**Clarity**

Section 3 is not clearly written, and I found it hard to follow and assess its correctness. Specifically:

- In Lines 173-174, is $R_{\ast}$ assumed to be $U_{\ast} \odot U_{\ast} - V_{\ast} \odot V_{\ast}$?
- In Lines 203-205, $\mathcal F$ is a set of loss functions. What is the exact meaning of "about the data"? Why is $R$ not considered in $\mathcal F$? Is a fixed $R$ being used here?
- In Lines 206-207, what is the definition of $\epsilon$-cover?
- In Lines 207-208, what are the mathematical definitions of the "average losses"?
- In Lines 210-213, it seems that here $R_{\ast}$ is fixed. Yet, it does not make sense to me because $R$ should depend on the transition matrix $T$ and the distributions $\mathbb D$ and $\tilde{\mathbb D}$. What is "ground truth" w.r.t. here?

**Significance**

The significance of the proposed method could be further enhanced through a more rigorous theoretical analysis (see above).

**Questions:**

Besides my questions listed above, here are my additional questions:

- For real-world noisy datasets, how did you get clean test labels?

- In Appendix C.3, what are symmetric noise and flip noise?

---

Minors:

- Eq. (20): $R^*$ ---> $R_*$.

**Limitations:**

I did not see where the authors discussed the limitations of the proposed method.

---

> ### Comment · Reviewer_T3zW · 2024-08-09
> **Maintain my initial rating**
>
> Since the authors did not post their responses, I maintain my initial rating.

---

### Official Review · Reviewer_dsrS · 2024-07-08

**Soundness:** 1
**Presentation:** 2
**Contribution:** 2
**Rating:** 3
**Confidence:** 3

**Summary:**

In noisy label learning problem, noise is often characterized by confusion matrix. In contrast to instance-independent noise, this work considers a setting where confusion matrices could be different for different samples. Under this setting, the authors proposed to use a global confusion matrix shared by all instances and a residual term for each instance to account for the different between instance-dependent confusion matrix and the global confusion matrix. For learning, an MLE loss combined with an implicit sparsity regularizer is optimized.

**Strengths:**

The work is tackling a challenging yet important setting in noisy label learning. The proposed model is a natural and intuitive extension to instance-independent confusion matrix as it allows for a wider range of noise. The proposed algorithm (TMR) is simple to implement, and demonstrated to be effective under synthetic and real-data experiments.

**Weaknesses:**

- Motivation for the use of sparsity regularizer is not clear. The authors does not discuss much on why the vector $\textbf{r}$, or matrix $\textbf{R}$ in their model should be sparse. They did point out in page 3, line 117 that the difference when using the global transition matrix and the instance-dependent transition matrix should be small. However, that is not sufficient to promote sparsity, as any other $l$-p ($l>1$) norm could have promoted that goal.
- The use of implicit regularizer is also not clear. And more importantly, since the output of $\textbf{T}^T P(\textbf{Y} | X) + \textbf{r}(X)$ is a probability vector, $\textbf{r}$(X) has to satisfy certain constraints. This is not discussed nor specified anywhere in the paper. And hence it is questionable how the parameterization of $\textbf{r}(X)$ could produce valid probability vector $\textbf{T}^T P(\textbf{Y} | X) + \textbf{r}(X)$.
- The analysis might contain flaw. Equation (14) is incorrect: $\widetilde{\textbf{Y}}$ is a matrix composing of one-hot vectors while the RHS is a matrix composing of probability vectors. The two are not equal in general. This equation seems to be the key step to motivate the objective to be analyzed in (17), and also the key step in the proof of Theorem 3.1 (page 15, line 534).
- The analysis is based on linear model which is not very realistic.

**Questions:**

- Should condition 1 be a design of the algorithm instead of a condition, since the learning rate should be in our control?
- In real-data experiments, should features extracted from self-supervised learning technique also be used for baselines, since it is applicable to most methods?
- How noisy is generated in synthetic setting?
- How does TMR combat a larger noise? Are there hyper-parameters we can tune in such situations? If yes, how were they selected in the experiment? If no, can you provide some intuition on how different noise level can be dealt with using the same algorithm configuration?

**Limitations:**

Yes

---

> ### Comment · Reviewer_dsrS · 2024-08-11
>
> Since the authors did not post their responses, I decide to maintain my initial rating.

---

### Official Review · Reviewer_17ZU · 2024-07-11

**Soundness:** 2
**Presentation:** 2
**Contribution:** 2
**Rating:** 3
**Confidence:** 4

**Summary:**

This paper introduces a method that supplements the traditional estimate of a class-dependent transition matrix, which is popular in label-noise learning. Traditional transition matrix methods are less effective for instance-dependent noise. To overcome the limitation, the proposed method adds a residual term such that it can extend the projection of a class-dependent T on label predictions to fit the true one as if we have an instance-dependent T. Theoretical analyses of the algorithm confirm its convergence and generalization properties under specific assumptions. Experimental results on various synthetic and real-world noisy datasets such as CIFAR-N and Clothing1M show the performance.

**Strengths:**

1. The performance is eye-catching.
2. The method is proposed with both theoretical analyses and experimental results.

**Weaknesses:**

1. The intuition of the proposed residual is not clear. For example, why a sparse structure is preferable in this problem? Why do u and v enable a sparse structure? Why is a Hadamard product employed? Why not simply use a vector u?
2. The theoretical part of the main paper is heavy but the outcome is not convincing. Specifically, there is a huge gap between Eq. (17) and Theorem 3.1.
3. The assumption in Eq. (7) is too strong.

**Questions:**

1. Is Theorem 3.1 based on infinite data or finite data?
2. It is not clear how the convergence is guaranteed. The critical results refer to another paper.
3. How do you guarantee the uniqueness of $\mathbf \theta^*$, $\mathbf T^*$, and $\mathbf R^*$? I believe it is hard to prove and simply assuming the uniqueness is too strong.

---

### Decision · Program_Chairs · 2024-09-25

**Decision:**

Reject

**Comment:**

This paper studies the problem of learning with instance-dependent noisy labels. the authors proposed to use a global confusion matrix shared by all instances and an additional residual term for each instance. For learning, an MLE loss combined with an implicit sparsity regularizer is optimized. The authors also provide a theoretical analysis of the proposed method. The strengths of the paper are theoretical analysis and simple and effective methods. However, weaknesses are a lack of novelty, insufficient motivation, missing baselines and less clarity in the mathematical formulation.  Three reviewers gave negative scores and one positive with lower confidence so the paper can not be accepted in this form. Therefore, it is suggested that the author carefully reorganize the entire paper following the suggestion provided by the review.